# Polyarenes Distribution in the Soil-Plant System of Reindeer Pastures in the Polar Urals

**Elena Shamrikova** [1,*] **, Evgeniia Yakovleva** [1]**, Dmitry Gabov** [1]**, Egor Zhangurov** [1]**, Michail Korolev** [1]
**and Elya Zazovskaya** [2]

[1]   Institute of Biology, Komi Science Center, Ural Branch, Russian Academy of Sciences,
     167982 Syktyvkar, Russia; kaleeva@ib.komisc.ru (E.Y.); gabov@ib.komisc.ru (D.G.); zhan.e@mail.ru (E.Z.);
     mihailkorolev1997@gmail.com (M.K.)
[2]   Institute of Geography of Russian Academy of Sciences, 119017 Moscow, Russia; zaszovsk@gmail.com
[*]   Correspondence: shamrikovaelena@yandex.ru

**Abstract:** Humic substances of soils consist of various organic compounds, including polycyclic aromatic hydrocarbons (PAHs). Soil as a fairly stable medium allows the correct use of polyarenes as markers of the humus formation process. Monitoring of the accumulation of PAHs as resistant organic toxicants is also necessary due to their carcinogenic and mutagenic properties. Natural plant resources serve as the feed base of northern reindeer husbandry. In this study, high-performance liquid chromatography in a gradient mode and gas chromatography-mass spectrometry methods were used to estimate the content of PAHs in mountain tundra and meadows of the Polar Urals (Russia). The accumulation of polyarenes in soils on carbonate rocks of the Bolshoi Paipudynsky ridge occurs mainly in the process of soil formation and largely depends on factors such as productivity of plant communities, the composition of standing biomass, the site's position in relief, the granulometric composition of soils, cryogenesis process and pyrogenesis. According to the set of polyarenes, their number, and ratio, the studied objects were classified into separate groups by discriminant analysis. The most typical representatives of pedogenic origin are naphthalene and phenanthrene. The accumulation features of dibenz[a,h]anthracene and benz[b]-, benz[k]fluoranthene, benz[ghi]perylene, and benz[a]perylene are shown. In mountain tundra landscapes, the characteristics of PAHs can be used to diagnose the intensity and direction of soil formation processes in general and humification in particular.

**Keywords:** PAHs; standing biomass; permafrost soils; coal particles; Ural Mountains; carbonate-rich materials

## 1. Introduction

Increased interest in the study of component composition of soil organic matter in recent years is caused by global climatic changes. An increase in temperatures and a change in snow accumulation trends that are especially pronounced in ecosystems of a cold humid climate, change the nature of humus formation. At the same time, the mountain soil cover of the Eurasian High Arctic has practically remained unstudied [1]. Due to better drainage and increased solar radiation, the soil organic matter of high-latitude mountain ecosystems can be more susceptible to transformation under the conditions of the increasing thermal pollution of the atmosphere compared to the soils of lowland landscapes. Soils that are formed on the products of weathering dense carbonate rocks remain one of the least studied in the Polar Urals.

The northern end of the Ural Mountains is actively used for private small-scale reindeer herding. Reindeer breeding and use is a vital agricultural branch in the Far North of the Russian Federation. Reindeer meat, liver, and kidneys are actively consumed by representatives of small indigenous peoples. However, a significant excess of the normative

indicators for dioxin content in deer liver was noted in most regions, including the Yamalo-Nenets Autonomous Area [2]. Northern ecosystems are prone to the accumulation of persistent organic pollutants due to the peculiarities of the climate, which prevents the decay of hazardous substances [3]. In this regard, the biomass composition of tundra plant species as the main food supply for deer is the focus of researchers' attention [4]. Among others, persistent organic toxicants include polyaromatic hydrocarbons (PAHs). There are data about the formation of dioxins and similar substances from PAHs in the course of their chlorination [5].

Polyarenes are the organic compounds of the benzene series that differ in the number of benzene rings and the peculiarities of their attachment. PAHs are highly mobile and dispersible in the biosphere. There are light 2–4 nuclear (naphthalene (NP), acenaphthene (ACE), fluorene (FL), phenanthrene (PHE), anthracene (ANT), fluoranthene (FLA), pyrene (PYR), benzo[a]anthracene (BaA), chrysene (CHR)) and heavy 5–6 nuclear PAH elements that are conventionally distinguished, including persistent organic pollutants (benzo[b]fluoranthene (BbF), benzo[k]fluoranthene (BkF), benzo[a]pyrene (BaP), dibenzo[a, h]anthracene (DahA), benzo[g, h, i]perylene (BghiP), indeno[1,2,3-c, d]pyrene (IcdP) [6–9].

The accumulation of PAHs in soils is associated with both organic matter transformation and incomplete combustion of organic materials (coal, oil, gasoline, wood, etc.) [10–12]. The low-molecular-weight PAHs are a part of the wax on the surface of leaves and insect cuticles and are produced in the digestive tract of termites [13–15]. The biological origin of naphthalene, phenanthrene, and perylene is indicated by their presence in plants, especially in tree trunks and termite nests [16]. However, forest fires of both modern and past geological epochs are recognised as the main natural source of polycyclic aromatic hydrocarbons in soils. Factual information about the role of paleofires on this issue is fragmentary [17–21].

The high resistance and hydrophobicity of polyarenes allow them to be adsorbed by organic matter and mineral matrices. In this regard, soils are the main landscape component that deposits PAHs [22], which makes it possible to use polyarenes as the markers of current and previous stages of soil formation. Bioaccumulation of PAHs in the soil-plant system occurs both due to endogenous and exogenous processes in the soil, as well as due to intracellular synthesis of PAHs [23].

In this regard, a detailed study of the accumulation and migration of PAHs as the components of soil organic matter on carbonate rocks of the Polar Urals is relevant and of scientific and practical interest.

The hypothesis of this study is that the accumulation of PAHs in northern mountain ecosystems will depend on the location of the site on the slope and the composition of its biomass and may be due to a number of factors: the processes of cryogenesis, weathering, and the granulometric composition of soils. The accumulation of PAHs in soils and plants will lead to their migration along the food chains of the Northern biogeocenoses.

The aim of this work is to reveal the regularities of PAH composition formation in soils on carbonate rocks that are developing in the northern part of the Bolshoi Paipudynsky ridge. Previously, the distribution features of organic and inorganic forms of carbon and nitrogen in soils for these objects, including their soluble fractions, were shown [1,24].

## 2. Materials and Methods

### 2.1. Site Description and Soil Sampling

The soil sampling area is located in the southern part of the circumpolar zone and covers the mountainous landscape zone of the Polar Urals. Outcrops of massive marbled Lower Devonian limestones are widely developed in this territory. Carbonate bedrock within the studied slope of northeastern exposure forms a thin cover of sediments that differ in granulometric composition. The climate of this region is humid continental characterised by sharp fluctuations in seasonal and daily temperatures. The amount of precipitation varies greatly depending on the height of mountain ranges and amounts to 600–800 mm per year [1]. Permafrost rocks are predominantly characterised by insular distribution.

The soils of a 1.5 km catena in the northern part of the Bolshoi Paipudynsky ridge were selected as research objects. The plots are located on a high-altitude profile from the root bank of the Razvilny Stream (plot 1, coordinates: 67°13′28.7″ N; 65°28′39.8″ E) to plateau in the slope upper part (plot 8, coordinates: 67°13′33.3″ N; 65°38′04.8″ E) (Figure 1). Soils of plots 1, 2, and 6–8 are formed on eluvial-deluvial deposits of carbonate rocks. The sections of plots 3–5 are laid on the concave part of a slope on re-deposited loose sediments that are sharply underlain by a massive platform. The field diagnostics of the studied soils and the determination of their classification position were performed according to the World Reference Base for Soil Resources [25].

The plant communities of selected sites differ significantly in composition, structure, and functional characteristics [1,24]. Groups are distinguished according to the above-ground biomass reserves of plant communities. The first is high-herb meadow (plot 7) and dwarf dryad moss tundra (plot 8) and it approximates 1600 g m$^{-2}$. Herb-shrub and herbal ecosystems are predominated by plants, the aboveground organs of which completely or mostly die off by the end of the growing season. The second group unites plant communities of plots 1–6, which have land biomass reserves of less than 200 g m$^{-2}$. The minimum values of productivity are characteristic of polygonal dryad tundra (plot 6) which forms under the eluvial conditions of the upper part of the piedmont ridge. Plot 6 has a cryogenic-spotted micro relief character with zones (plot 6, spot) not covered with vegetation (Figure 1). On plots 1, 3, and 6, vegetation is represented by sparse groups that are dominated by Dryas octopetala. The total projective cover ranges from 0 to 60%. Plots 2, 4, 5, 7, and 8 are characterised by more developed vegetation cover. The total projective coverage is 100%.

Post-fire charcoal inclusions of various sizes are clearly diagnosed in the morphological structure of plot 6 at a depth of 2–35 cm. The intra-profile distribution of carbonaceous particles in this section indicates a local and rather powerful type of fire. Subsequent erosional processes of redeposition (flat washout of upper soil layers during the period of active snow melting and a large amount of precipitation during summer) led to further migration of carbonaceous particles down the slope. The most striking example of this is the presence of buried horizons with a large amount of coal in the middle profile part of plots 4 and 5. Fragments of charcoal differ from dark-colored mineral particles of humus horizons in black colour (2.5Y N2/black; 10YR 2/1 black), light reflectance, strength, shape, and size (Figure 2). Carbonaceous soil particles of plot 5 from a depth of 10–15 cm were analysed.

8. Calcaric Folic Gleysol (Skeletic)    7. Calcaric Stagnosol Humic Skeletic    6. Leptic Skeletic Calcaric Regosol    5. Mollic Leptic Calcaric Stagnosol Skeletic    4. Leptic Skeletic Calcaric Regosol    3. Folic Mollic Calcaric Leptosol Humic    2. Mollic Calcaric Stagnosol Skeletic    1. Mollic Calcaric Stagnosol Skeletic

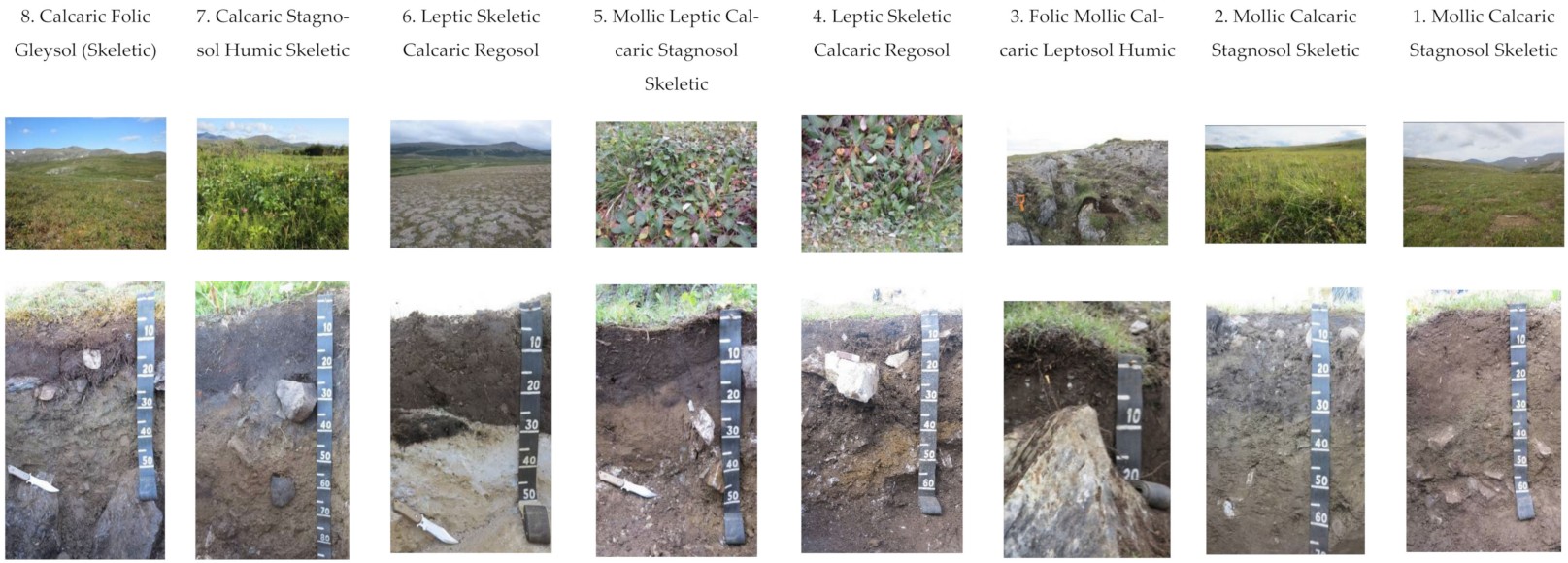

**Figure 1.** Soil profiles and vegetation (numeral—No. of plot/soil profile).

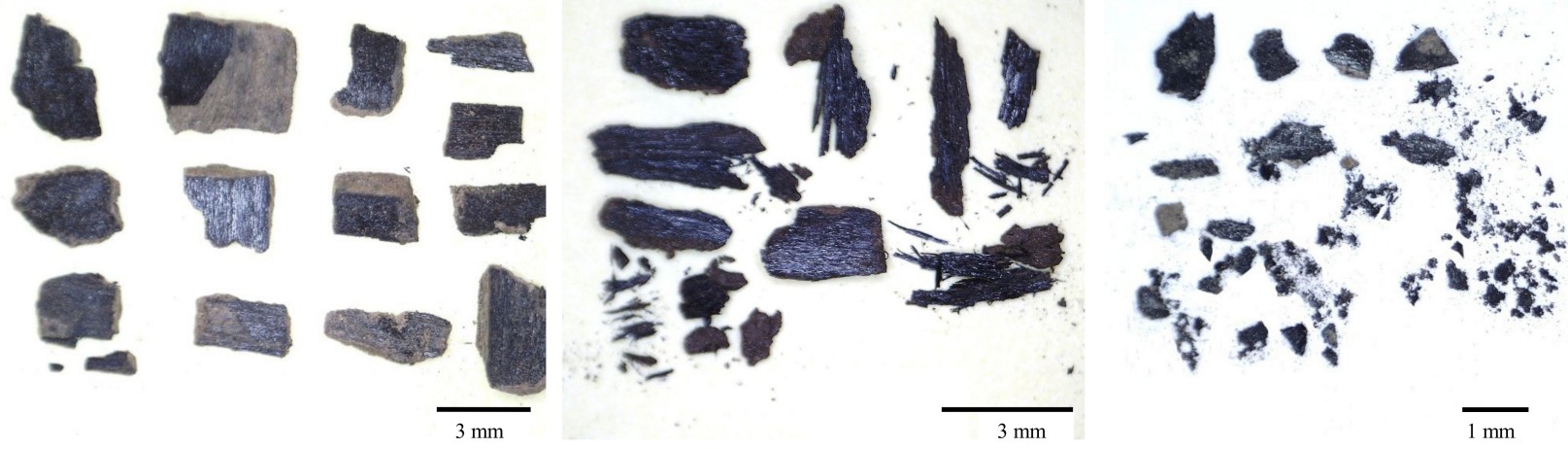

**Figure 2.** Carbonaceous particles morphology.

## 2.2. Chemical Analysis of Soils

Chemical analyses were performed in the Chromatography Collective Use Center of the Institute of Biology, Federal Research Center, Komi Research Center of the Ural Branch of the Russian Academy of Sciences. A total of 15 PAHs were analysed, including naphthalene (NP), acenaphthene (ACE), fluorene (FL), phenanthrene (PHE), anthracene (ANT), fluoranthene (FLA), pyrene (PYR), benzo[a]anthracene (BaA), chrysene (CHR), benzo[b]fluoranthene (BbF), benzo[k]fluoranthene (BkF), benzo[a]pyrene (BaP), dibenzo[a,h]anthracene (DahA), benzo[g,h,i]perylene (BghiP), and indeno[1,2,3-c,d]pyrene (IcdP). The extraction of PAHs from the soils and plants was made on a Dionex™ ASE™ 350 Accelerated Solvent Extractor (Dionex/Thermo Fisher Scientific™, Waltham, USA). Weighed portions (1 g) of soil (or plant) were placed into extraction cells; the extraction was performed three times with a mixture of methylene chloride: acetone (1:1) at 100 °C. Then, the extracts were concentrated using a Kuderna-Danish concentrator at the thermostat temperature of 65 °C and the solvent was replaced with hexane. The obtained samples of 2 cm$^3$ in volume were purified from organic impurities according to the US EPA purification method 3660c (1996c) by column chromatography using activated aluminum oxide (Brockmann II grade) (Fluka, cat. no. 06300, particle size 0.05–0.15 mm, activated at 600 °C for 4 h, and partially deactivated with 3% $H_2O$). A 0.5 cm layer of sodium sulfate was added to the top of the column. The hexane: methylene chloride (4:1) mixture (30 cm$^3$) was applied as an eluent.

The eluates were concentrated using a Kuderna-Danish concentrator at the thermostat temperature of 90 °C to a volume of 1–2 cm$^3$. Then, 3 cm$^3$ of acetonitrile were added to them and these mixtures were evaporated at 90 °C until the complete removal of hexane. Finally, the collected solution was preserved in a freezer at −20 °C prior to analysis. The relative errors of determination (at P = 0.95, ±δ, %) depended on the measurement range and varied within 16, 36, 50 for NP, 20, 22, 40 for ACE, 18, 26, 40 for FL, 20, 22, 50 for PHE, 18, 24, 50 for ANT, 18, 38, 46 for FLA and PYR, 20, 26, 42 for BaA, 22, 28, 52 for CHR, 22, 26, 42 for BbF, 18, 24, 48 for BkF, 18, 24, 50 for BaP, 20, 30, 48 for DahA, and 22, 24, 44 for BghiP and IcdP (measurement ranges 1000–2000 µg/kg, 100–1000 µg/kg, 5–100 µg/kg, respectively).

A standard mixture of 15 PAHs (Supelco EPA 610 PAHs) with concentrations of each component in the range of 100–2000 µg cm$^{-3}$ was used to prepare the 5 standard PAH solutions with concentrations of each component in the range of 5–1000 ng cm$^{-3}$. A standard mixture, blank, and sample duplicate were run in each sample batch (max. 10 samples) for checking quantification, contamination, peak identification, method precision, and accuracy. No target compounds were detected in blanks.

Total concentrations of carbon and nitrogen ($C_{tot}$ and $N_{tot}$) in the soil samples were measured by using a Carlo Erba EA-1100 CHN analyser. Concentrations of carbonates were determined in soil samples with $pH_{H2O}$ > 6.8 by the volumetric method using a calcimeter 08.53 Eijkelkamp (SA07, USA–Netherlands) [25].

The charcoal radiocarbon dates were obtained by accelerator mass-spectrometry (IGANAMS). Sample preparation, separation of the datable fraction, graphitization, and pressing on a target was performed in the Laboratory of Radiocarbon Dating and Electronic Microscopy of the Institute of Geography RAS, Moscow (lab code IGANAMS). Graphitization was performed using an AGE-3 graphitization system [26]. The AGE-3 uses a Vario Micro Cube elemental analyzer coupled to an Isoprime PrecisION IRMS (Elementar). Graphite $^{14}C/^{13}C$ ratios were measured by applying the CAIS 0.5 MeV accelerator mass spectrometer at the Center for Applied Isotope Studies, University of Georgia, USA. The sample ratios were compared to the ratio measured from the Oxalic Acid II (NBS SRM 4990C). All $^{14}C$ ages were calculated by applying the δ13C value of −25%. The conventional radiocarbon ages were calibrated (2σ standard deviation) applying the CALIB Rev 8.2 program, with the IntCal20 calibration data set.

The values of pH in $H_2O$ were determined potentiometrically with a glass electrode ES-11.7 (Akvilon, Russia) using soil with the following characteristics: solute ratio of 1:2.5

for mineral and 1:25 for organic horizons, respectively. Simple regression analyses were performed using Statistica 6.0 software packages.

## 3. Results

All the studied objects have revealed the presence of 14 polyarene structures, including naphthalene, fluorene, phenanthrene, anthracene, fluoranthene, pyrene, chrysene, benz[a]anthracene, benz[b]fluoranthene, benz[k]fluoranthene, benz[a]pyrene, benz[ghi]perylene, dibenz[a,h]anthracene, and acenaphthene. It should be noted that acenaphthene was found only in coal samples (Figure 3). The content of PAHs in the standing plant biomass in plots varies in the range of 35–110, in coal–770, and in soils–6–190 μg kg$^{-1}$ (Table 1).

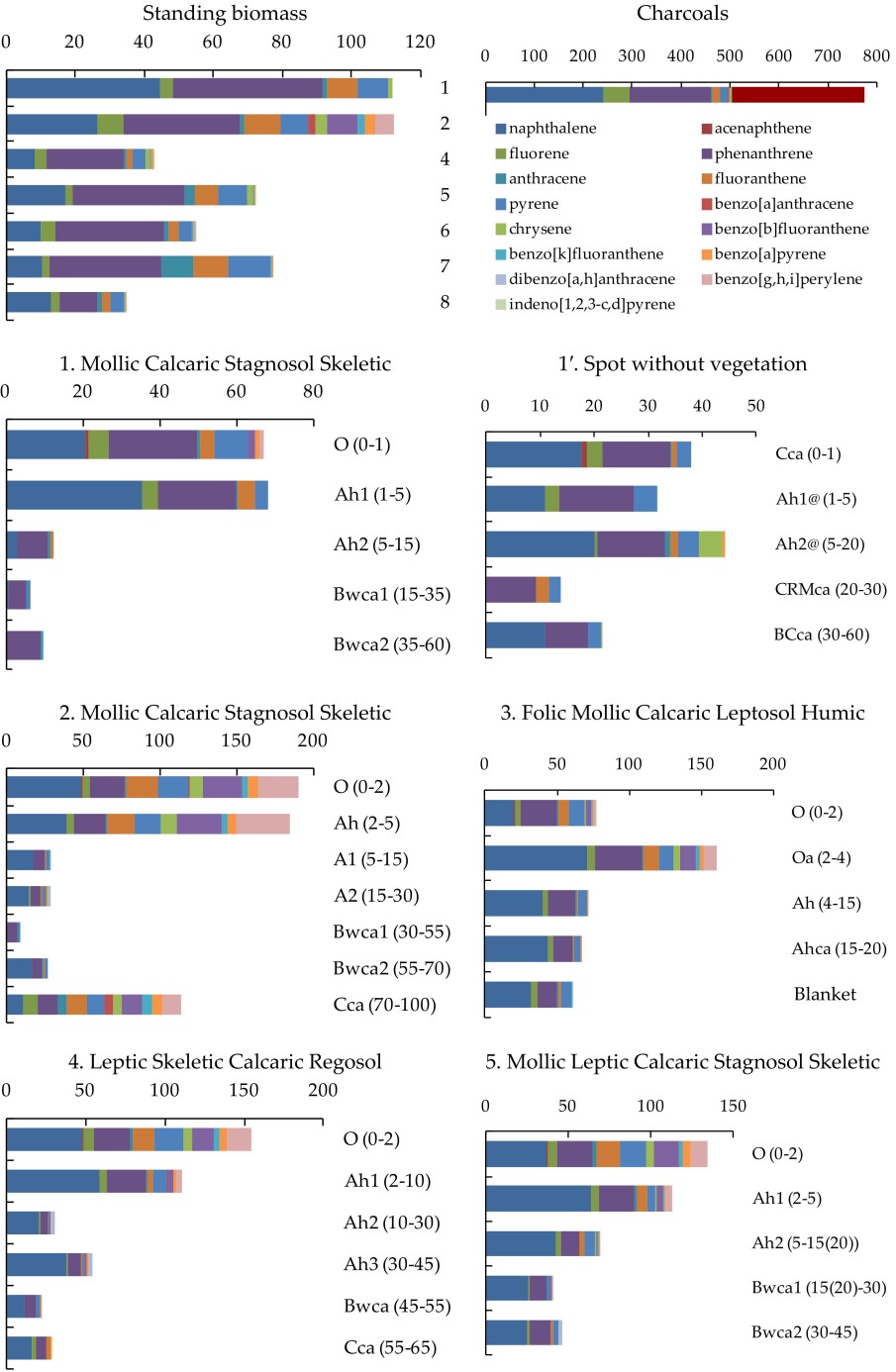

**Figure 3.** *Cont.*

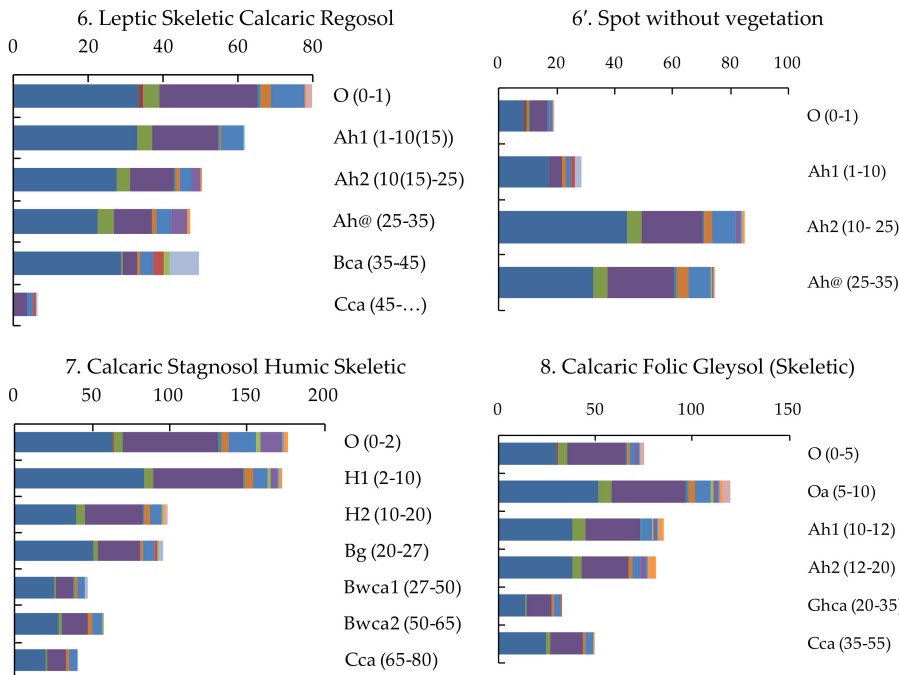

**Figure 3.** PAH content of standing biomass, carbonaceous particles, and soils ($\mu$g kg$^{-1}$).

**Table 1.** Chemical characteristics of soils.

| Soil Type | Horizon | Depth | pH$_{H2O}$ | C$_{tot}$ | C$_{inorg}$ | Ntot | C$_{PAH}$ [1] |
|---|---|---|---|---|---|---|---|
| | | cm | | g kg$^{-1}$ | | | $\mu$g kg$^{-1}$ |
| | Standing Biomass | – [2] | | 443.0 | 0.0 [3] | 11.8 | 105.4 |
| 1. Mollic Calcaric Stagnosol Skeletic | O | 0–1 | 6.67 | 147.0 | | 9.4 | 62.2 |
| | Ah1 | 1–5 | 6.99 | 33.0 | | 2.4 | 64.1 |
| | Ah2 | 5–15 | 7.33 | 33.0 | 0.6 | 2.3 | 11.5 |
| | Bwca1 | 15–35 | 7.72 | 11.8 | 6.1 | 0.7 | 6.1 |
| | Bwca2 | 35–60 | 7.90 | 7.6 | 5.2 | 0.5 | 8.9 |
| 1′. Spot without Vegetation | Cca | 0–1 | 7.84 | 1.4 | 0.2 | 0.1 | 34.8 |
| | Ah1@ | 1-5 | 7.87 | 1.1 | 0.0 | 0.1 | 29.8 |
| | Ah2@ | 5–20 | 7.57 | 1.0 | 0.0 | 0.1 | 41.5 |
| | CRMca | 20–30 | 7.89 | 0.7 | 0.2 | 0.1 | 13.1 |
| | BCca | 30–60 | 8.05 | 0.9 | 0.6 | 0.0 | 20.3 |
| 2. Mollic Calcaric Stagnosol Skeletic | Standing Biomass | – | | 421.0 | 0.0 | 17.5 | 106.1 |
| | O | 0–2 | 7.00 | 243.0 | | 17.2 | 179.6 |
| | Ah | 2–5 | 6.71 | 132.0 | 0.8 | 10.9 | 174.7 |
| | A1 | 5–15 | 7.47 | 43.0 | 4.9 | 2.9 | 26.1 |
| | A2 | 15–30 | 7.62 | 44.0 | 9.1 | 3.1 | 26.6 |
| | Bwca1 | 30–55 | 7.92 | 9.2 | 6.5 | 0.4 | 7.9 |
| | Bwca2 | 55–70 | 7.95 | 3.5 | 1.1 | 0.4 | 25.2 |
| | Cca | 70–100 | 8.10 | 4.4 | 4.1 | 0.2 | 107.9 |
| 3. Folic Mollic Calcaric Leptosol Humic | Standing Biomass | – | | 485.0 | 0.0 | 19.8 | – |
| | O | 0–2 | 7.22 | 241.0 | 1.4 | 12.9 | 72.1 |
| | Oa | 2–4 | 7.40 | 188.0 | 2.5 | 11.7 | 151.1 |
| | Ah | 4–15 | 7.50 | 63.0 | 5.5 | 5.2 | 66.5 |
| | Ahca | 15–20 | 7.69 | 57.0 | 4.4 | 4.9 | 63.1 |
| | Blanket [4] | | 7.70 | 67.0 | 10.6 | 5.3 | 56.0 |

**Table 1.** *Cont.*

| Soil Type | Horizon | Depth | pH$_{H2O}$ | C$_{tot}$ | C$_{inorg}$ | Ntot | C$_{PAH}$ [1] |
|---|---|---|---|---|---|---|---|
| | | cm | | g kg$^{-1}$ | | | µg kg$^{-1}$ |
| 4. Leptic Skeletic Calcaric Regosol | Standing Biomass | | 6.1 | 42.0 | 0.0 | 1.9 | 40.2 |
| | O | 0–2 | 7.0 | 25.7 | 0.0 | 1.8 | 144.6 |
| | Ah1 | 2–10 | 7.3 | 8.9 | 0.0 | 0.9 | 104.3 |
| | Ah2 | 10–30 | 7.5 | 6.3 | 1.6 | 0.5 | 28.5 |
| | Ah3 | 30–45 | 7.6 | 5.7 | 1.5 | 0.4 | 50.7 |
| | Bwca | 45–55 | 7.8 | 6.7 | 6.1 | 0.1 | 20.1 |
| | Cca | 55–65 | 7.7 | 6.0 | 1.6 | 0.4 | 26.1 |
| 5. Mollic Leptic Calcaric Stagnosol Skeletic | Standing Biomass | – | | 440.0 | | 18.9 | 67.8 |
| | O | 0–2 | 6.90 | 302.0 | | 19.6 | 125.8 |
| | Ah1 | 2–5 | 6.59 | 110.0 | 0.0 | 9.8 | 106.2 |
| | Ah2 | 5–15(20) | 6.97 | 76.0 | | 6.9 | 65.3 |
| | Bwca1 | 15(20)–30 | 7.40 | 18.0 | | 1.5 | 38.4 |
| | Bwca2 | 30–45 | 7.73 | 16.0 | 6.6 | 1.3 | 43.3 |
| 6. Leptic Skeletic Calcaric Regosol | Standing Biomass | – | | 420.0 | 0.0 | 9.4 | 51.8 |
| | O | 0–1 | 7.57 | 236.0 | 70.3 | 8.7 | 74.1 |
| | Ah1 | 1–10(15) | 8.04 | 116.0 | 81.1 | 3.2 | 57.8 |
| | Ah2 | 10(15)–25 | 8.08 | 124.0 | 50.2 | 5.4 | 47.4 |
| | Ah@ | 25–35 | 7.98 | 121.0 | 44.0 | 6.2 | 44.3 |
| | Bca | 35–45 | 8.32 | 118.0 | 103.8 | 0.0 | 46.6 |
| | Cca | 45–. . . | 8.50 | 118.0 | 122.4 | | 5.7 |
| 6′. Spot without Vegetation | O | 0–1 | 7.76 | 110.0 | 91.3 | 2.2 | 17.2 |
| | Ah1 | 1–10 | 7.93 | 112.0 | 93.1 | 2.2 | 26.7 |
| | Ah2 | 10–25 | 7.89 | 115.0 | 77.4 | 4.0 | 79.7 |
| | Ah@ | 25–35 | 8.00 | 126.0 | 67.2 | 4.7 | 70.1 |
| 7. Calcaric Stagnosol Humic Skeletic | Standing Biomass | – | | 405.0 | | 23.3 | 72.6 |
| | O | 0–2 | 5.58 | 331.0 | | 25.4 | 164.9 |
| | H1 | 2–10 | 6.30 | 243.0 | 0.0 | 17.1 | 162.1 |
| | H2 | 10–20 | 6.49 | 167.0 | | 12.0 | 92.5 |
| | Bg | 20–27 | 6.61 | 21.0 | | 1.6 | 90.1 |
| | Bwca1 | 27–50 | 7.61 | 16.0 | 7.0 | 0.6 | 44.6 |
| | Bwca2 | 50–65 | 7.97 | 22.0 | 24.6 | 0.4 | 53.4 |
| | Cca | 65–80 | 8.04 | 21.0 | 24.4 | 0.4 | 37.9 |
| 8. Calcaric Folic Gleysol (Skeletic) | Standing Biomass | – | | 410.0 | | 8.5 | 32.7 |
| | O | 0–5 | 6.41 | 384.0 | | 17.3 | 69.8 |
| | Oa | 5–10 | 6.77 | 345.0 | 0.0 | 16.0 | 112.4 |
| | Ah1 | 10–12 | 6.91 | 198.0 | | 12.9 | 80.2 |
| | Ah2 | 12–20 | 6.95 | 156.0 | | 10.3 | 76.4 |
| | Ghca | 20–35 | 7.72 | 27.0 | 5.5 | 1.6 | 30.5 |
| | Cca | 35–55 | 7.99 | 16.0 | 7.6 | 0.8 | 46.5 |

[1] PAH carbon, [2] not determined, [3] below the determination limit, [4] redeposited material from the slope.

Light structures dominate PAH biomass. In most areas, their contribution to the total amount of PAHs is 97–100%. The exception is biomass composition in plot 2, where the proportion of light PAHs decreases to 80%. Maximum concentrations were found for phenanthrene and naphthalene, the contribution of which is 20–50 and 20–40% of the total PAH content in biomass. The biomass of plots 1 and 2 is characterised by the maximum accumulation of PAHs. The lower content is noted in the biomass of plots 4–7.

The studied soils are characterized by a distinct differentiation into genetic horizons, which rapidly effervesce under the influence of 10% HCl throughout the entire profile. The upper (surface) horizons are represented by litter-peaty (O and Oao), consisting of a mechanical mixture of organic residues of various degrees of decomposition with mineral components. The humus horizons lying below are dark (black) in color and are diagnosed as Ah (Ah1; Ah2; Ah3) with the accumulation of humified organic matter closely associated

with the mineral part of the soil. In the middle and lower parts of the profile, the number of fragments of carbonate rocks increases sharply and passes into large blocks of rock. In the soil under the meadow (plot 7) formed in a depression, the H1-H2 horizons are formed in the upper part of the profile—dark gray, moist, structureless, consisting of decomposed plant remains that have lost their original structure. The accumulation of snow and the watering of this area during the snowmelt period cause the formation of gley processes in the middle part of the profile (horizon Bg). The maximum content of rocks represented by limestones is typical for sections 6 and 8 (70–80% of the horizon volume), in other sections, they prevail only from a depth of 35–40 cm.

All profiles, except for the soil from plot 6, are characterised by the accumulation of PAHs in organogenic horizons. In all studied soils, as well as in plants, light PAHs dominate. Their share out of the total amount of polyarenes varies from 76% to 100%, depending on a soil horizon and plot. PAHs are mainly represented by naphthalene and phenanthrene. For plot 2, as well as for plants, there is a decrease in the proportion of light PAHs to 60–67% in the upper horizons. This fact is largely related to the influence of PAHs composition in vegetation. In soils, as in biomass, an increase in banz[a]pyrene and benz[ghi]perylene was revealed.

A two to four-fold excess of the PAH content in organogenic horizons of soils under vegetation was revealed in comparison with the soils of bare spots from plots 1 and 6. The soils from plots 2 and 7 differ in the maximum content of PAHs. For them, a 2-fold or even a higher excess of the PAH content in O(0-2) and H1(2-10) horizons was revealed in comparison with biomass. Similar patterns have been identified for plots 4 and 5. At a depth of 5–15 cm of plot 5, the presence of a fragmentary buried horizon with coals was noticed. Polyaromatic spectra of pyrogenic particles are also mainly represented by light PAHs—acenaphthene, phenanthrene, and naphthalene (35, 31, and 21%, respectively). Soils of plots 3 and 8 are characterised by a low PAH content.

## 4. Discussion

In the course of detailed morphological studies in the upper horizons of plot 6 (1–10 cm and 10–25 cm), a significant number of small coals with dimensions of 1 mm or less was revealed. Cryogenic microrelief of this site is pronounced (polygons without vegetation account for up to 70% of the area). Active cryoturbation processes (including physical disintegration processes) largely determine the dispersion of carbonaceous material. In the lower horizon at a depth of 25–35 cm, the content of large carbonaceous particles increases significantly (from 3–5 mm to 10–12 mm). According to a number of authors [21,24,27], this indicates a local type of fire. The presence of a powerful buried layer of coals (with dimensions up to 5–6 mm or more) in karst craters (located lower down the slope from plots 6 and 5) at a depth of 35–50 cm demonstrates the profile of buried soil with a distinct differentiation into genetic horizons that are formed as a result of erosion-decomposition processes after fires.

Based on the results of radiocarbon dating of coal particles from plot 6, the age was determined—6340 ± 30 years ago (in the karst funnel lower down the slope, the age of coal particles is 5340 ± 30 years ago), which corresponds to the late Atlantic climatic optimum of the Holocene. Scarce literature data on palynology of the Polar Urals peatlands that are geographically close to the area of our work indicate that the climate was much warmer than the modern one [17,28]. The main tree species are Picea, Larix, and Betula with a small admixture of broad-leaved species [29]. Forest vegetation in the foothills and on the plains moved within the limits of the modern tundra at 200–400 km (to the shores of the Barents Sea). In the spore-pollen spectra, charcoal particles that are formed as a result of a series of paleofires were clearly diagnosed [28].

The studied sites are located at a considerable distance from industrial enterprises and urban centres. Therefore, the current anthropogenic impact on the composition of PAHs in plants and upper soil layers is limited. According to the composition assessment of ice cores and snow in remote regions, the global transfer of PAHs is insignificant. Minor contents of

fluoranthene and pyrene (0.1–4.7 µg kg$^{-1}$) and benzo[g,h,i]perylene (0.05–0.18 µg kg$^{-1}$) were found in meltwater [30–33].

### 4.1. Standing Biomass

The dominance of phenanthrene and naphthalene in plant and soil samples that was revealed by us was shown earlier. These are the compounds that, as a rule, prevail in natural phytocenoses plants of tundra with a low content or absence of high molecular weight PAHs [19,29,30]. The values of total PAH content in the soils that were found in studied plots are, in general, typical for the soils of background tundra plots [34].

The increased content of PAHs in the biomass of plots 1 and 2 can be explained by the mechanical transfer of organic matter (including carbonaceous particles) from the plateau surface to the accumulative positions of the landscape. Material movement is carried out by aeolian processes, by the shift of snow masses that are capable of carrying organic residues with them from the slope, as well as by washing off the upper soil layers during the period of active snow melting and a large amount of precipitation. This fact is indicated by streams' beds that are running from the ridge top to its foot. The species composition of these sites is dominated by moss vegetation that is capable of a hyper-accumulation of PAHs due to the year-round growing season and active absorption of atmospheric deposition by the entire surface [35].

Dryads and shrubs with dense leaves poorly accumulate PAHs from outside. This leads to a decrease in the content of PAHs in the biomass of plots 4 and 7. The plant surface morphology, specifically the cuticular wax micro-structure, is an essential factor that regulates the deposition, distribution, and penetration of organic pollutants into and across plant cuticles. A decrease in the cuticle PAH content with an increase in wax in it has been shown many times [36]. In addition, the low ability of shrubs and grasses to accumulate PAHs is determined by a short vegetation season [11,34].

No increased PAH content was found in the biomass of plot 8, although mosses are represented significantly in the composition of the site species. The shrubs that form a canopy could be an obstacle to polyarenes' entry into the surface of lower layer plants. In addition, plot 8 is located in the upper slope part, and PAHs on the moss surface could be washed out and migrate down the slope. Up to 20% of polyarenes in natural tundra phytocenoses are concentrated on moss surfaces and do not penetrate into plants [34].

### 4.2. Soils

PAHs are adsorbed by soil organic matter and can be chemically stable for a long time in a conserved state [37–39]. This fact explains the concentration of polyarenes in the upper organogenic horizons of most studied soils. Cryoturbation processes that occur at the top of a hill lead to the inversion of polyarene content, as well as the overall content of organic carbon and nitrogen in soils [1].

Plots 1 and 6 have a cryogenic-spotted micro relief character. Plots under vegetation are combined with bare spots. The absence of vegetation in both cases affects the composition of upper horizon polyarenes. The PAH content in the two upper soil horizons of plot 1 is two times lower than in the presence of vegetation due to the absence of vegetation. This is connected with the significant content of polyarenes found in the plant biomass of this plot, which is two times higher than the content of PAHs in soil (mainly due to naphthalene and phenanthrene). The absence of heavy polyarenes in the biomass of plot 1 and their presence in the organogenic horizon indicates that these compounds were formed in the course of soil-forming processes. These include microbiological decomposition of vegetation, for example, through sequential hydroxylation of aromatic rings. Naphthalene and phenanthrene, which dominate the biomass of this plot, are capable of decomposing in the soil to simple compounds that are not related to PAHs during soil formation. The next reaction is the synthesis of humic substances from organic compounds that are present in plants, including pentacyclic terpenes, aromatic structures, as well as structures with diene or polyene bonds in aliphatic hydrocarbon chains and lipids [38,40,41]. In addition,

phenanthrene that is accumulated by mosses might not enter the soil profile in significant amounts. Mosses, in comparison with herbaceous and woody species, are characterised by a low capacity for decomposition [42,43]. In general, PAHs can both serve as structural units during the formation of humic substances and be formed during their destruction [44].

Close correlations between the content of individual PAHs in the biomass and organic horizon O (0–1) of the soil of plot 1–r = 0.97 (n = 13, $p < 0.05$, $r_{cr} = 0.56$) were revealed. The connection density slightly decreases r = 0.95 (n = 13, $p < 0.05$, $r_{cr} = 0.56$) in the sub-litter horizon. PAHs can migrate to this horizon from the overlying one; therefore, they were represented only by the most soluble light polyarenes. A rise in the number of rings in PAH molecules increases their molecular weight and hydrophobic properties [45]. The correlation coefficients of PAHs composition in biomass and soils naturally decrease with depth and in Ah horizon (5–15) amount to r = 0.87 (n = 13, $p < 0.05$, $r_{cr} = 0.56$).

Similar patterns were obtained for the soil of plot 6. The mass fractions of PAHs in the upper organogenic horizon of a section under vegetation are four times higher than under a cryogenic spot. Polygonal dryad tundra is characterised by minimal values of biological productivity; significant concentrations of PAHs in biomass were not revealed. High values of the correlation coefficients of individual PAHs content in biomass and upper organogenic horizon of the soil from this plot were detected: r = 0.79 (n = 13, $p < 0.05$, $r_{cr} = 0.56$), which successively decrease with depth in the underlying strata r = 0.57 (n = 13, $p < 0.05$, $r_{cr} = 0.56$).

Heavy polyarenes were found in the lower part of plot 6. No such migration was observed in plot 1. It is explained by the fact that in the first area mineral horizons are loams [1], where migration of PAHs is hindered in comparison to the sandy horizons of plot 6, where heavy PAHs could penetrate to great depths, which is consistent with [46]. It resulted from the increased sorption capacity of soils that are formed on loamy rocks. Clay minerals, along with soil organic matter, are considered the two most important factors that affect the sorption of PAHs in soil [47]. At the same time, the intensity of PAH sorption by organic matter is higher than by clay minerals [48]. In plot 6, most heavy PAHs are present only in the lower profile part, which may be caused by cryoturbation processes that are common for this area. Thus, our results confirm the previously drawn conclusion that the low content of organic matter and colloid particles in a soil horizon are the factors that promote the migration of PAHs deep into a profile [49].

The maximum mass fractions of soils PAH in plots 7 and 2 are explained by their presence in depressions, which contributes to PAHs accumulation. The second factor is the type of plant community. The high-grass meadow in plot 7 is a highly productive community that forms in positions that are the snowiest in winter and most humidified in summer. Aboveground plant organs mostly die off by the end of a growing season, which leads to the active formation of an organogenic horizon—the main depot of PAHs. Biomass of herbaceous species rapidly decomposes, which leads to the supply of additional amounts of both light and heavy PAH structures to the upper soil horizons. Herbaceous plants in comparison to mosses contain large amounts of lignin [40,50,51], which is the main source of condensed aromatic molecules [52]. The latter is a prerequisite for a more than two-fold excess of PAHs in O(0-2) and H1(2-10) compared to biomass. The content of PAHs (both heavy and light) decreases steadily depthward. The concentration of individual biomass polyarenes and the upper organogenic soil horizon is also interrelated –r = 0.78 (n = 13, $p < 0.05$, $r_{cr} = 0.56$), where the correlation coefficient decreases with depth.

In the soil of plot 2, which forms under a less productive herb sedge moss tundra community, PAHs are concentrated in the two upper horizons O(0-2), Ah(2-5), where their amount is 1.7 times higher than the biomass. An increase in the proportion of heavy PAHs in total mass (50–66%) is noted in the upper horizons as well as in plants. This may be due to a slower decay of mosses compared to shrubs and herbaceous plants. This leaves PAH composition unchanged for a longer time. In addition, the decomposition of lignin-rich sedges, which are widely represented in the first layer, could also lead to the formation of significant amounts of heavy PAHs in the upper profile part.

The intra-profile migration in plot 2 is less intense in comparison to plot 7 due to the lower relief location of the first. Probably, in the upland areas, a more intense washout regime is formed, which promotes the active migration of polyarenes. In addition, plot 2 is also characterised by a peak of polyarenes accumulation in the lower part of the profile—BC (70–100). A sharp increase in the total carbon content was also diagnosed at the same depth (Table 1). Apparently, the PAH pool is associated with the historical formation of this soil, since at this depth there was an increase in the content of all PAH structures under the absence of heavy polyarenes in the three overlying layers.

Soils of plots 4 and 5 that are located in the middle slope part and formed under a herb willow tundra community are characterised by similar patterns of PAH accumulation. The PAH content in upper horizons is also two to three times higher than in the biomass. Formation of PAHs in soil results from the active decomposition of shrub-herbaceous vegetation that is enriched with lignin. An increased content of high molecular weight polyarenes such as benz[b]fluoranthene, benz[k]fluoranthene, benz[a]pyrene, and benz[ghi]perylene, which are in total up to 32% of the overall PAH mass, was noted. Polyarenes also uniformly migrate to the underlying horizons.

The presence of a fragmentary buried horizon with coals at a depth of 5–15 cm in plot 5 that is laid down in the transit-accumulative conditions of the landscape indicates the allochthonous (introduced) nature of post-fire coals that were displaced from the epicentre of combustion (plot 6). In the course of prolonged redeposition (most likely, washout by water flows), the transformation of the parent material may have taken place, contributing to reliable "encapsulation" of PAHs that are contained in coal particles. Chemical stability ("conservation") of PAHs for a long time due to their adsorption by organic matter in the soil was indicated earlier [53–55].

The absence of the PAH accumulation peak in the 5–15 cm horizon of plot 5 indicates that PAHs in coal did not migrate as they were reliably bound in coal particles. According to Campos et al. (2019), fire immediately causes an increase in PAHs (mainly light) in the topsoil. After four months, the mass fraction of PAHs significantly decreases [17]. There is an opinion [14,17] that PAHs can migrate from the soil into the atmosphere and plants along the soil profile during the first year after the fire. In our case, it is possible that the most inert part of coals, which was "washed" (leached) at the site of the fire, was exposed to redeposition. A decrease in PAHs extractability with their long-term presence in soils was recorded earlier [44–46].

The presence of acenaphthene only in coal samples indicates that this compound is a combustion product and does not migrate into soils and plants, as it is fixed in coals. It is assumed that at the first time after the passage of fire, this hydrocarbon could be washed out from the soil, but in the process of microbiological decomposition, it apparently disintegrated into simple substances. This fact explains its absence in soils and plants for the study period. After the passage of fire, leaching into the underlying horizons and down the slope likely took place. It is this process that could explain the detection of the increased concentrations of polyarenes at the slope foot of plot 2 at a depth of 70–100 cm.

The low content of PAHs in soils at plots 3 and 8 is related to the fact that these soils were formed in the upper slope part under various types of plant communities that are characterised by dryad presence. These factors were likely of significant importance in the formation of the polyarene spectra composition of these soils. The influence of the former has been shown by a number of researchers [35,36]. The upper horizons of both plots are represented by weakly decomposed or undecomposed biomass, which determines the low content of PAHs. The correlation coefficient between the composition of PAHs in biomass and O-horizon (0–5) for plot 8 was $r = 0.97$ ($n = 13$, $p < 0.05$, $r_{cr} = 0.56$). In more decomposed parts of the organic soil layer, the content of PAHs increases due to the formation of light and heavy structures and their downward migration. Polyarenes, including high molecular weight ones, are capable of migrating to fairly large depths [17].

The discriminant analysis data (Figure 4) made it possible to differentiate the studied objects into separate groups according to the set of polyarenes, their number, and ratio.

The first group includes objects with a low PAH content (6–50 μg kg$^{-1}$). The light representatives (NP, FL, PHE, PYR) account for up to 85% of the total mass. In addition, only in this group the accumulation of DahA (5–16%)—a heavy polyarene—is expressed. The group is composed of middle humus horizons that are located at a depth of 15–45 cm, soils of plots 2, 4, 5, as well as the horizons of plot 6 deeper from the 35 cm mark and the spot layer without vegetation at a depth of 2–10 cm. As it was indicated above, in this profile without vegetation, deep horizons are displaced to the surface by cryogenesis processes. The most pronounced composition specificity of this group is the characteristic of the Bca (35–45) horizon at plot 6, where the proportion of DahA in the total mass of polyarenes is maximal. Probably, this terrane is the main source of dibenz[a, h]anthracene that is formed as a result of a local fire. Further, DahA was mechanically moved along the slope to the transit-accumulative position (plot 2).

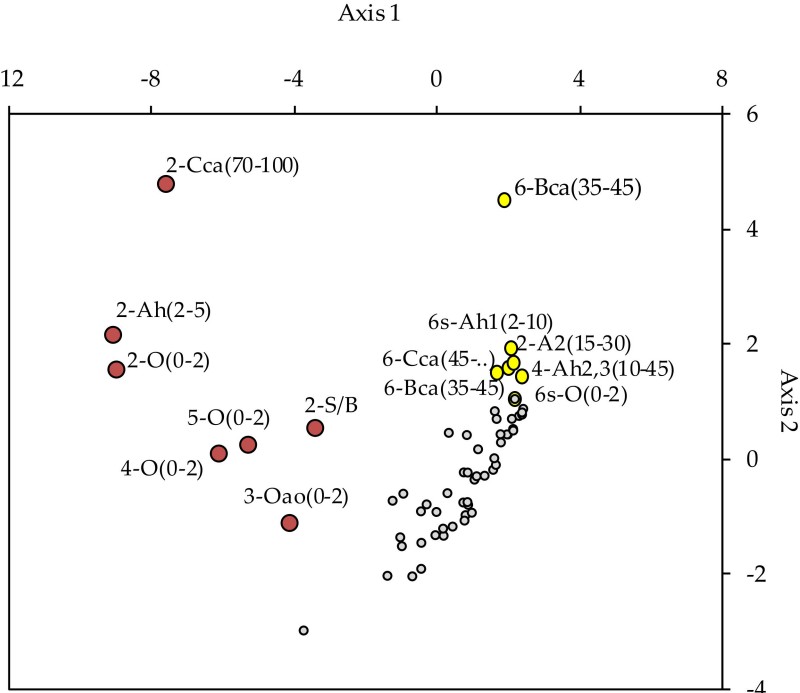

**Figure 4.** Ordinary diagram (principal components analysis). Diagnostic signs—the content of individual PAHs (1–3 groups of horizons, explanations in the text). Note: digit—the section number, S/B—standing biomass.

The specificity of the second group is greater accumulation of PAHs—110–150 μg kg$^{-1}$, as well as the presence (only here) of heavy BbF and BghiP (7–12% each). The total share of NP, FL, PHE, PYR, and FLA in the total mass of polyarenes is 40–70%. This set is represented by organogenic soil horizons of plots 3–5, biomass, and two upper and buried soil layers of plot 2. The predominant presence of BbF and BghiP in the upper horizons indicates a later occurrence of these polyarenes in soils compared to DahA. The record-high total share of BbF and BghiP (35%) falls on the Ah(2–5) horizon of plot 2. The most specific in this group is the buried horizon BC(70–100) of plot 2, where heavy polyarenes make an equal fraction of total mass. In addition to BbF and BghiP, BkF and BaP are also present here. The significant specificity of this horizon composition reflects the complex history of its formation. It is possible that this horizon was buried in the previous stages of soil formation.

The third group which is the most extensive includes the objects that are similar in relative composition of polyarenes. The amount of PAHs in them is mainly provided by the presence of five components—NP, FL, PHE, PYR, and FLA, which are light PAHs. These objects are distinguished only by a wide variation in the total amount of compounds—6–180 μg kg$^{-1}$.

It should be noted that these heavy polyarenes are carcinogenic. BaP has particular toxicity and mutagenic effects and is classified as the I (highest) hazard class in the Russian Federation. The maximum concentration value of BghiP in soils is 20 μg kg$^{-1}$ (GN 2.1.7.2041-06). It should also be mentioned that in the biomass and individual horizons of plot 2, the content of this carcinogen is significant (up to 6.3 μg kg$^{-1}$). Bioaccumulation of organic toxicants in the soil-plant-animal system is a risk for the basic function of the traditional Far Northern sector of the agro-industrial complex. According to Russian regulations, the content of benzaperene in meat and meat-containing products can not exceed 1 μg kg$^{-1}$ [2].

## 5. Conclusions

Polyaromatic hydrocarbons that possess high stability, hydrophobicity, and adsorption capacity are preserved in soils for a long time. Studies of these compound groups in a soil cover are relevant due to both their indicative features and their pronounced carcinogenic and mutagenic effects. Pollution with PAHs—persistent organic pollutants—is damaging vulnerable ecosystems in the Far North. The investigated area is used as forage for reindeer pastures.

Standing biomass has a significant effect on the content of polyarenes in soils. The highest content of polyarenes is observed in the profiles that are formed under a tall grass meadow. This is facilitated by maximum productivity of the plant community, the annual supply of standing plant organs to a soil surface, and enrichment of herbaceous plant residues with lignin.

Pyrogeneses are an important factor of soil formation, causing activation of exogenous geomorphological processes in the previous cycles of soil formation (erosion of landscapes, redeposition, and subsequent migration of carbonaceous material). When analysing the content and distribution of PAHs, it is certainly necessary to take into account the complex paleogeographic context of sedimentation in glacial and interglacial epochs, as well as complicated climatic fluctuations in the Late Holocene and vegetation dynamics (forest-tundra). At the present stage of soil formation, the distribution of PAHs in soils is also associated with the course of cryogenic processes. The content of polyarenes in the upper horizons of soils of cryogenic spots is several times lower than in the soils of corresponding areas under vegetation. This fact is associated both with the absence of additional input of PAHs from plants and with the processes of weathering (physical disintegration), to which cryogenic spots were subjected to a greater extent. Permafrost phenomena caused by maximum freezing of soils at the top of the ridge determines the inversion of PAHs distribution in the profile, with their highest content in the lower horizons of a soil section.

The peculiarities of a leaching regime, which depends on the granulometric composition of soils and the nature of rubble, determine the intra-profile migration of polyarenes in soils. Significant movement along the profile is expressed in sandy loam and structureless light loamy soils. The burial of PAHs at a depth of 70–100 cm occurred due to the historical migration processes of organic compounds after a local fire.

Regular monitoring of the accumulation of polyarene and plant products of the territory in soils is required to control their possible entry through food chains into the body of animals and humans. In the future, it is planned to study the composition of deer muscle tissues for the content of PAHs.

**Author Contributions:** Conceptualization, E.S. and E.Y.; methodology, E.Y., D.G., E.Z. (Egor Zhangurov), and E.Z. (Elya Zazovskaya); software, E.S. and E.Y.; validation, E.Z. (Egor Zhangurov); formal analysis, D.G. and E.Z. (Elya Zazovskaya); investigation, E.S., E.Z. (Egor Zhangurov), and M.K.; resources, E.Y. and D.G.; data curation, E.S. and E.Y.; writing—original draft preparation, E.S. and E.Y; writing—review and editing, E.S., E.Y., D.G., E.Z. (Egor Zhangurov), and M.K.; visualization, E.S.; supervision, E.Z. (Egor Zhangurov), E.Z. (Elya Zazovskaya), and M.K.; project administration, E.S.; funding acquisition, E.S. and M.K. All authors have read and agreed to the published version of the manuscript.

**Funding:** The reported study was funded by the federal budget (1021051101421-1-1.6.1) and RFBR grant (no. 20-04-00445a).

**Institutional Review Board Statement:** Not applicable.

**Informed Consent Statement:** Not applicable.

**Data Availability Statement:** Not applicable.

**Acknowledgments:** Authors thank Evgeniia Shamrikova for help.

**Conflicts of Interest:** The authors declare no conflict of interest. The funders had no role in the design of the study; in the collection, analyses, or interpretation of data; in the writing of the manuscript; or in the decision to publish the results.

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
