# Peer review of "Polyarenes Distribution in the Soil-Plant System of Reindeer Pastures in the Polar Urals"

_agronomy, doi:10.3390/agronomy12020372_

Round 1

Reviewer 1 Report

In general, the work "Polyarenes distribution in the soil-plant system of reindeer pastures in the Polar Urals" presents scientific merits and brings important studies to the scientific community outside the domains of the Russian Federation.
However, some aspects should be noted by the authors (these suggestions are placed in the pdf document, in the form of comments):

- Introduction:
1) The hypothesis clearly stated in the Introduction is missing (comment on line 78).

- Materials and Methods:
2) The authors need to inform in the Material and Methods, which was the Soil Classification system used in the work - this is fundamental (comment on line 83).
3) The analytical results of the particle size distribution (Texture) are missing - comment on line 123.

- Results:
4) Line 199 (Table 1): It remains to describe or characterize the horizons of each type of soil studied. The nomenclature alone does not help in reading the work - professionals who are not part of the field of Pedology or Physical Geography of Soil will have great difficulty in understanding the results presented.

- Discussion:
5) Comment added between lines 318-329: Soil texture results are lacking, for evaluated horizons. Also, information about the type of clay would be important [silicate clays? Montmorillonite (Smectites), Vermiculites, Micas...???] present in the soils of these plots. Please discriminate what is the dominant mineralogy in each soil selected in the M&M. Otherwise, all claims and comparisons with these clay minerals will be invalidated.

- Conclusions:
6) Conclusions 3 and 4 need to be revised and rewritten - they need to be synthesized. It's more for results and discussion, rather than conclusions!!!

Note: For authors to observe all suggestions in the pdf document. sent by the reviewer.

Author Response

Dear colleagues!  The team of authors expresses their deep gratitude to the reviewer for their attentive attitude to our manuscript.

The authors agree with all the comments of the reviewer. The authors believe that the correction of the manuscript did help to improve the presentation of our results. 

- Introduction:
1) The hypothesis clearly stated in the Introduction is missing (comment on line 78).

Authors. We have added a hypothesis to the Introduction.

The hypothesis of the study is that the accumulation of PAHs in northern mountain ecosystems will depend on the location of the site on the slope and the composition of its bomass and may be due to a number of factors: the processes of cryogenesis, weathering, and the granulometric composition of soils. The accumulation of PAHs in soils and plants will lead to their migration along the food chains of the Northern biogeocenoses.

- Materials and Methods:
2) The authors need to inform in the Material and Methods, which was the Soil Classification system used in the work - this is fundamental (comment on line 83).

Authors. We agree, this is our omission. We have added a link in the text and bibliograph).

The field diagnostics of the studied soils and the determination of their classification position were performed according to the World Reference Base for Soil Resources [IUSS Working Group WRB, 2015].

IUSS Working Group WRB. 2015. World Reference Base for Soil Resources 2014. International soil classification system for naming soils and creating legends for soil maps. World Soil Resources Reports â„– 106. FAO, Rome.. 

3) The analytical results of the particle size distribution (Texture) are missing - comment on line 123.

Authors. We agree. Indeed, unfortunately, there are no data on particle size distribution. The granulometric composition of soils was determined by field methods and on the basis of tactile sensations. Due to the high content of carbonates in the soil fine earth, analytical results of particle size distribution were not performed.

Soil description guide. Fourth edition, revised and enlarged Rome: Food and Agriculture Organization of the United Nations (FAO). 2012, 101p.

- Results:
4) Line 199 (Table 1): It remains to describe or characterize the horizons of each type of soil studied. The nomenclature alone does not help in reading the work - professionals who are not part of the field of Pedology or Physical Geography of Soil will have great difficulty in understanding the results presented.

Yes, we are agree. The fact that we did not take into account this aspect is an omission. Now we have added the characteristics of the horizons of different soil types. In conjunction with the named classification of soils, it will be easier for specialists who are not related to the field of soil science or the physical geography of soils to understand the presented results.

The studied soils are characterized by a distinct differentiation into genetic horizons, which rapidly effervesce under the influence of 10% HCl throughout the entire profile. The upper (surface) horizons are represented by litter-peaty (O and Oao), consisting of a mechanical mixture of organic residues of various degrees of decomposition with mineral components. The humus horizons lying below are dark (black) in color and are diagnosed as Ah (Ah1; Ah2; Ah3) with the accumulation of humified organic matter closely associated with the mineral part of the soil. In the middle and lower parts of the profile, the number of fragments of carbonate rocks increases sharply and passes into large blocks of rock. In the soil under the meadow (section 7), formed in a depression, the H1-H2 horizons are formed in the upper part of the profile - dark gray, moist, structureless, consisting of decomposed plant remains that have lost their original structure. The accumulation of snow and the watering of this area during the snowmelt period causes the formation of gley processes in the middle part of the profile (horizon Bg). The maximum content of rocks represented by limestones is typical for sections 6 and 8 (70-80% of the horizon volume), in other sections they prevail only from a depth of 35-40 cm.

- Discussion:
5) Comment added between lines 318-329: Soil texture results are lacking, for evaluated horizons. Also, information about the type of clay would be important [silicate clays? Montmorillonite (Smectites), Vermiculites, Micas...???] present in the soils of these plots. Please discriminate what is the dominant mineralogy in each soil selected in the M&M. Otherwise, all claims and comparisons with these clay minerals will be invalidated.

Authors. Of course, we agree with the remark about the need to provide information on the mineralogical composition of the clay fraction. However, to date, such a highly informative and complex analysis is not possible and is beyond the scope of this article. The assumption is made on the basis of literary data.

- Conclusions:

6) Conclusions 3 and 4 need to be revised and rewritten - they need to be synthesized. It's more for results and discussion, rather than conclusions!!!

Authors. Yes, we are agree. Indeed, the reduction of conclusions 3 and 4 made the conclusion more concise.

…The investigated area is used as forage reindeer pastures.

Standing biomass has a significant effect on the content of polyarenes in soils. The highest content of polyarenes is observed in the profiles that are formed under a tall grass meadow. This is facilitated by maximum productivity of the plant community, annual supply of standing plant organs to a soil surface, and enrichment of herbaceous plant residues with lignin.

Note: For authors to observe all suggestions in the pdf document. sent by the reviewer. Yes, we have carefully corrected all the comments in pdf format.

Additionally, we have corrected the English. Corrections are highlighted in yellow.

Sincerely, authors.

Reviewer 2 Report

Shamrikova et al. investigated the accumulation of polyarenes in soils on carbonate rocks. Influencing factors such as productivity of plant communities, composition of standing biomass, site position in relief, granulometric composition of soils, cryogenesis process and pyrogenesis have been discussed. The findings in this work improves the understanding about the mechanisms governing PAHs accumulation in soil formation, and has important implication to humus formation process. The manuscript is written well, but many experimental conditions are not clearly presented. I suggest that the authors address the mentioned issues and be more outspoken concerning the uncertainty of the obtained numerical values.

  1. Line 20, an “and” should be removed.
  2. More detailed results are supposed to be presented in the abstract, rather than conclusions.
  3. Lines 46-47, “Northern ecosystems are prone to accumulating persistent organic pollutants, since they have the necessary characteristics of climate and food chains”. This sentence is confused, please explain what kind of necessary characteristics.
  4. Lines 51-60, please provide references.
  5. Lines 149-151, it seems like the concentration results have large variation. In addition, the standard concentration of PAH was prepared as 100-2000 µg/cm3 (line 153), which is much higher than the concentration of PAH detected in soils (6.1-179.6 µg/kg, table 1). Please explain the validity and reliability of your data.
  6. Please provide detailed information on horizons (O, Ah, Bca, Cca…).
  7. Figure 1. plot 4 is missed.
  8. Figure 2. please use English in the legend. Please explain the abbreviations in the figure such as O, Ah1, Ah2…, so the figure can “stand alone”.
  9. Format of reference list should be reorganized.

Author Response

Dear colleagues!  The team of authors expresses their deep gratitude to the reviewer for their attentive attitude to our manuscript.

The authors agree with all the comments of the reviewer. The authors believe that the correction of the manuscript did help to improve the presentation of our results.

  1. Line 20, an “and” should be removed.

Authors. Thank you, your comment has been noted.

  1. More detailed results are supposed to be presented in the abstract, rather than conclusions.

Authors. Yes, we are agree. Indeed, we have shortened the conclusion, it has become more concise.

  1. Lines 46-47, “Northern ecosystems are prone to accumulating persistent organic pollutants, since they have the necessary characteristics of climate and food chains”. This sentence is confused, please explain what kind of necessary characteristics.

Authors. We agree that this proposal needs to be corrected. It is more about the specifics of the climate. Low temperatures prevent the decomposition of opiate compounds. We have corrected this proposal.

Northern ecosystems are prone to the accumulation of persistent organic pollutants due to the peculiarities of the climate, which prevents the decay of hazardous substances.

  1. Lines 51-60, please provide references.

Authors. Links has been added to the manuscript.

  1. Lines 149-151, it seems like the concentration results have large variation. In addition, the standard concentration of PAH was prepared as 100-2000 µg/cm3 (line 153), which is much higher than the concentration of PAH detected in soils (6.1-179.6 µg/kg, table 1). Please explain the validity and reliability of your data.

Authors. Indeed, this is an important clarification.

The relative errors of determination have large variation because their depended on the measurement range. Information about relative errors and the measurement ranges were added in manuscript.

The concentration of standard reference material Supelco EPA 610 PAHs. The calibration standards with concentrations of each component in the range of 5-1000 ng cm-3 were prepared from this standard reference material. This information has been added to the manuscript (lines 163-169). Measurements of PAHs concentrations in soils were carried out within the range of calibration standards.

  1. Please provide detailed information on horizons (O, Ah, Bca, Cca…).

Abbreviations such as O, Ah1, Ah2 … denote the indexes of soil horizons according to the WRB classification. We have added a link to this post. We have added information about the horizons in the Manuscrip.

  1. Figure 1. plot 4 is missed.

Agreed, the soil profile from site 4 has been added to Figure 1.

  1. Figure 3. please use English in the legend. Please explain the abbreviations in the figure such as O, Ah1, Ah2…, so the figure can “stand alone”.

Authors. The legend for the figure has been corrected. Thank you, sorry omission!

Abbreviations such as O, Ah1, Ah2 … denote the indexes of soil horizons according to the WRB classification. We have added a link to this post. We have added information about the horizons in the Manuscript.

  1. Format of reference list should be reorganized.

Authors. Thank you, we have taken this note into account.

Additionally, we have corrected the English. Corrections are highlighted in yellow.

Sincerely, authors.
